# Are There Changes in Inequalities in Injuries? A Review of Evidence in the WHO European Region

**DOI:** 10.3390/ijerph16040653

**Published:** 2019-02-22

**Authors:** Mathilde Sengoelge, Merel Leithaus, Matthias Braubach, Lucie Laflamme

**Affiliations:** 1Department of Public Health Sciences, Karolinska Institutet, Widerströmska Huset, Tomtebodavägen 18 A, 171 77 Stockholm, Sweden; lucie.laflamme@ki.se; 2Department of International Health, Maastricht University, Universiteitssingel 40, 6229 ET Maastricht, The Netherlands; merel@leithaus.net; 3WHO European Centre for Environment and Health, Platz der Vereinten Nationen 1, D-53113 Bonn, Germany; braubachm@who.int

**Keywords:** unintentional injuries, health inequalities, country-level differences, Europe, road traffic, falls, burns, poisonings

## Abstract

Decreases in injury rates globally and in Europe in the past decades, although encouraging, may mask previously reported social inequalities between and within countries that persist or even increase. European research on this issue has not been systematically reviewed, which is the aim of this article. Between and within-country studies from the WHO European Region that investigate changes in social inequalities in injuries over time or in recent decades were sought in PubMed, Scopus, and Web of Science. Of the 27 studies retained, seven were cross-country and 20 were country-specific. Twelve reported changes in inequalities over time and the remaining 15 shed light on other aspects of inequalities. A substantial downward trend in injuries is reported for all causes and cause-specific ones—alongside persisting inequalities between countries and, in a majority of studies, within countries. Studies investigate diverse questions in different population groups. Depending on the social measure and injury outcome considered, many report inequalities in injuries albeit to a varying degree. Despite the downward trends in risk levels, relative social inequalities in injuries remain a persisting public health issue in the European Region.

## 1. Introduction

Globally, injuries account for nearly 1.7 times the number of fatalities that result from HIV/AIDS, tuberculosis, and malaria combined [1]. As injuries are highly preventable [1], evidence-based prevention measures are promoted in a variety of reviews and policy documents targeting either specific groups, such as children and adolescents [2], or specific the major injury causes like road-traffic injuries [3], falls [4], drownings [5], or burns [6]. In general terms, the most effective preventive measures typically rest on the notion of “built-in safety” pertaining to how physical environments (e.g., that of home, road, or work) or of products (e.g., clothes, toys, vehicles) are designed [7]. Safety can also be enhanced through the promotion, more or less coercive, of safe practices or behaviours (e.g., smoke detectors, pool fencing), the success of which relies on enforcement measures, for example speed cameras. Even tertiary prevention measures may significantly help to reduce the burden of injuries, for example through improved access to trauma care and emergency medical services.

It is in part due to the diffusion and implementation of this prevention knowledge at large scale that remarkable downward trends in injury mortality and morbidity observed globally and in Europe [8] in recent decades have been attributed [9]. From 1990 to 2013 for instance, DALY rates for all injuries declined by 31% according to the Global Burden of Disease data, with declines varying by injury cause: in drowning a 52% decline, burns 47%, poisonings 44%, falls 21% and road traffic injuries 16% [9]. In the WHO European Region, child injury mortality rates have declined by 29% since 2000; 58% reduction in poisonings, 33% in road traffic and only 9% in falls [8]. Yet, these encouraging trends are not automatically reflected in all countries and may mask the maintenance (or even rise) of inequalities between and within countries.

Indeed, injuries may be considered as a fundamental cause of health inequalities as the material, economic or social means required to protect oneself (or one’s offspring) from injuries may not be available to all [10]. Countries with low implementation of promoting and reinforcing safety policies and measures tend to place the responsibility of prevention on local and individual initiatives, whereby injury prevention may receive lower priority due to other local competing needs such as housing, mobility or food security [11]. Over the years, social inequalities in injuries has been reported in numerous reviews, demonstrating that social inequalities in injuries are common but not consistent, as their magnitude varies according to the injury cause, injury severity and population group considered. The most recent WHO 2009 review focused on all injuries and the evidence showed strong associations with individual (and area-based) material deprivation. People from low socioeconomic status and from less affluent areas tend to die by injury to a greater extent than others [12]. Other reviews were target specific, by age group [13,14,15,16,17,18], injury cause, such as road traffic [19,20], falls [21], burns [22,23,24], or housing and neighbourhood conditions [25,26]. From among those reviews, only a few are specific to the European context, which is the focus of the present review, and to the best of our knowledge only one study [27] reviewed the literature on trends in social inequalities in injuries. This review from 2012 focused on child mortality from unintentional injuries and reported on the notion that overall reduction in mortality over time masked rising socioeconomic and ethnic inequalities; in particular road traffic-related injuries [27].

Given that the scientific literature on trends on social inequalities in injuries in the WHO European Region is scattered and has not been the subject of a more recent and broader review, the primary aim of this study is to shed light on the state of knowledge concerning changes over time in social inequality in injuries in the European Region. A secondary aim is to complement this description with an updated review of studies from the Region on social inequalities in injuries in injuries that are not concerned with changes over time but that present data more recent than those presented in the former review [12].

## 2. Materials and Methods

### 2.1. Search Strategy

Relevant studies on injuries, referring to damage to the body produced by energy exchanges that have sudden discernible effects [28], were searched in three electronic databases: PubMed, Scopus and Web of Science using the combination of keywords provided in Table 1.

### 2.2. Inclusion and Exclusion Criteria

Published studies were included for the literature review when they met the following criteria: (1) examined injury inequalities within the Region over time or providing other data published after the most recent WHO injury inequality review in 2009 [12]; (2) measured inequality at the individual level based on at least one of the ‘PROGRESS’ dimensions [29], area level (e.g., area deprivation index) or country level (e.g., gross national income, income inequality); (3) considered injuries of all causes aggregated or any of the major causes of injury burden that have been previously associated with inequalities: road traffic, drowning, poisoning, falls and burns; (4) severity of injury indicated as mortality, morbidity or both. Studies were excluded for either of the following: (1) publication not peer reviewed; (2) publication date before 2010; (3) publication in another language than English; (4) injury outcome data older than 2005; (5) focus on health care outcomes, quality of life, well-being after injury; (6) no available full text.

### 2.3. Literature Screening

The studies were screened according to the Preferred Reporting Items for Systematic Reviews and Meta-Analysis (PRISMA) flow diagram (Figure 1) [30]. According to the preset inclusion and exclusion criteria, title and abstract screening were performed by two of the authors. The authors then reviewed the 62 full texts and identified eligible studies, followed by the data extraction. If the opinions of the two authors were not uniform, it was resolved by discussing or referring to the opinions of a third author.

### 2.4. Data Extraction and Synthesis

The following information was extracted from each study included: research question, setting/population, design, whether the study reported results by age group or by sex, what measures of social position were used, how the data were treated and the significance of the results assessed. Also extracted were the main findings of the studies and their implications on the role of the environment on injury inequalities.

A narrative synthesis was performed of the study results stratified into two groups: (1) studies investigating changes over time; and (2) studies not investigating changes over time published after the 2009 WHO inequality review [12]. For each set of studies, a synthesized table was organized based on whether the studies were cross- or within-country and investigated all injuries aggregated or specific causes from among those retained for the review. If a study selected included injury causes (or injury types) not retained for the review, e.g., sports-related injury, these data were not included.

## 3. Results

After screening 1274 records a total of 27 studies [8,31,32,33,34,35,36,37,38,39,40,41,42,43,44,45,46,47,48,49,50,51,52,53,54,55,56] met the criteria for inclusion, as shown in Figure 1. Seven were cross-country and 20 country-specific. Twelve reported changes in inequalities over time and the remaining 15 on other aspects of inequalities. Most were at the area-level using an area deprivation index as a social inequality dimension, followed by education, and ethnicity (three studies from one study material). No studies reported information on religion, social capital or sexual orientation.

### 3.1. Studies of Changes over Time

Table 2 summarizes the 12 studies that report changes in the rate and distribution of injuries over time, organized based on whether they are cross-country studies (two studies) or within-country ones (ten studies).

#### 3.1.1. Cross-Country Studies over Time

The two cross-country studies were on childhood injuries in the European Region and investigated variations between high income countries (HICs) and low- and middle-income countries (LMICs) from 2000 to 2011 [31] and 2000 to 2015 [8]. Both studies based on very similar data report widening inequalities in unintentional injuries in spite of substantial decreases in absolute terms. For example, the rate ratio of unintentional injuries increased by 41% (*p* < 0.001), mostly due to the rate of road traffic injuries decreasing more rapidly in HICs (130% higher, *p* < 0.001) compared to LMICs [8].

#### 3.1.2. Within-Country Studies over Time

The second set of ten studies report changes within countries for all injuries aggregated (*n* = 4) followed by road traffic injuries (*n* = 4) and two studies on other injury mechanisms of interest [8,32,33,34,35,36,37,38,39,40]. They vary considerably as regards the research question and study group and reveal mixed results. Regarding all injuries, Regidor et al. [33] assessed differences in injury mortality between 1970 and 2010 comparing the richest and poorest Spanish provinces and found no association. Focusing on the adult population, Pekkala et al. [32] and Strand et al. [35] studied respectively changes in long-term occupation-related sickness absence due to injuries in Finland between 2005 and 2014 (age 25–64 years) and education-based all injury mortality between 1960 and 2010 in Norway (age 45–74 years). The former found that no significant change occurred for absolute differences in sickness absence but that relative differences narrowed [32] and the latter found that injury mortality among lower educated males and females remained systematically higher than that of males and females with higher education (slope of index inequality: 61 in men, 20 in women) [35]. A recent study on paediatric injuries in the United Kingdom considered changes in children having two or more injuries requiring medical attention in 1980 and 2012 [34]. When assessing household income-based differences no injury gradients were found when comparing the two time periods.

For the four studies on road traffic injuries over time, one is national from Israel (a member state of the WHO Regional Office for Europe and thus also covered by the systematic review) [36] and three are city-specific and area-based; one from Glasgow, Scotland [37] and two from London, England [38,39]. Comparing Arabs and Jews during the period 2003–2011, Magid et al. [36] found a mortality and morbidity reduction in both groups but Arabs were consistently at higher risk compared to Jews especially for the very young (rate ratio 8.81 for 0–4 age group) and older persons (rate ratio 3.08 for 60–64 age group). The study from Glasgow [37] and two from London [38,39] investigated the difference that a change in the traffic environment had introduced in road traffic injury risk level and in their social distribution. Olsen et al. [37] found that the introduction of a new motorway had not had any influence on the social patterning of injuries and that they remained more frequent in more deprived areas (annual proportion 30.6 in most deprived versus 1.0 in the least deprived in 2014). For their part, Steinbach and colleagues found that a reduced speed limit (20 mph zone) had similar reductions in road traffic injuries across quintiles of social deprivation. Yet because of the higher number of road traffic casualties in deprived areas, the socioeconomic inequalities have widened over time from 1987 to 2006 [38]. Finally, focusing on child pedestrian injuries Steinbach et al. studied the influence of group density and ethnicity from 2001 to 2011 and reported a significant decrease in injury risk for all three ethnic groups ‘White’, ‘Asian’ and ‘Black’ during this time period [39].

The two last studies in Table 2 considered social differences in poisoning or burn and poisoning injuries combined among children in the UK aged 0–4 years [40] and aged 10–17 years [41]. Orton and colleagues. Ref [40] found that between 1990 and 2009 burns and poisonings among the very young declined in children from all income quintiles but that strong socioeconomic inequalities persisted when comparing most to least deprived areas (28% of poisonings and 30% of burns attributable to deprivation). In contrast, Tyrrell and colleagues [42] found an increase in poisonings in adolescents during a similar time period (1992 and 2012) and a strong socioeconomic gradient (adjusted IRR = 2.83 in 1992–1996 and IRR = 2.63 in 2007–2012).

### 3.2. Other Studies

Table 3 presents studies reporting social inequalities in injuries in Europe in more recent years, stratified first according to whether they were cross-country [41,42,43,44,45] or within-country [47,48,49,50,51,52,53,54,55,56]. The studies that considered more than one cause are inserted in each section of the table for which they provide results. They were grouped in three categories related to the injury cause investigated: all injuries (four studies), road traffic injuries (four studies) and other injuries (four studies), of which one on falls [55] and three studying both burns and poisonings [50,51,56].

#### 3.2.1. Cross-Country Studies

The five cross-country studies investigated various populations; two on male and female adults [42,44], one sex-specific on people of all ages [42] and two on children [45,46]. Regarding all injuries, Gallo et al. [42] studied education level of adults and mortality differences across a range of cause-specific conditions using data from nine countries, either as a whole or represented by some cities; in men, they found that all injury mortality for those with highest education was 44% lower than for those with lowest education (IIR = 0.56, CI: 0.35–0.90 crude) and the association was not significantly affected (adjusted IRR = 0.61, CI: 0.38–0.98) after adjusting for one or multiple risk behaviours (smoking status, alcohol consumption, leisure physical activity, fruit and vegetable) or risk factors (body mass index measured at recruitment). In women, no such association was found and this was not influenced when taking the same risk factors into consideration. Gotsen and colleagues [43] investigated inequality differentials based on small area deprivation in cities of various numbers from 15 countries and grouped these into four different geographic areas of Europe; for men, they found a positive association between area deprivation and all injuries in the majority of cities/regions apart from Central eastern cities; the largest (relative) differences between men from the most and least deprived areas were in Stockholm (Sweden) (RR = 1.27, CI: 1.22–1.31) in the North. The studies on child injury mortality from Sengoelge and colleagues are based on national-level data from 26 European countries [45,46]. They investigated the associations—and interactions—between-country-level economic disparity (considering income inequality and gross domestic product (GDP) and housing conditions with child injury mortality, and whether the associations differ for different age groups. Combined, the studies show that all three measures were significantly associated with child injury mortality (income inequality r = 2.05, GDP r = −6.55, and housing strain r = 5.94) [45,46]. The associations are quite similar when looking at specific age groups (1–4, 5–9, 10–14 years) (47) and housing strain partially modified the associations between income inequality and GDP with all injury mortality [46].

Three of the cross-country studies considered road traffic-injury mortality. Two assessed its gender-specific social distribution in adults, either by age and area-level deprivation [43] or education [44]. The third is a sub-analysis in the study from Sengoelge and colleagues presented above [45]. Mackenbach et al. [44] found a higher mortality risk among the ‘low’ vs. ‘high’ educated adults in 19 European populations, and higher for men (median RR 2.06) than for women (median RR 1.26). When studying area-level deprivation, Gotsen et al. [43] demonstrated a significant positive association between deprivation and road traffic mortality in six European cities for men (highest differences in Stockholm (Sweden), RR = 1.18, CI: 1.07–1.29), but no association in women in the cities studied. Sengoelge and colleagues [46] found that the association between-country-level income inequality and child road traffic injury mortality was fully mediated by housing strain (r = −2.25, CI: −3.20–−1.31).

Gotsen et al. [43] and Mackenback et al. [44] also investigated falls. The latter demonstrated a higher mortality risk among the ‘low’ vs. ‘high’ educated adults which was similar for both sexes (median RR 1.84 men, median RR 1.55 women) which contrasts with Gotsen et al. [43] who found a positive association between deprivation and falls in four cities for men (highest differences in Lisbon (Portugal), Southern Europe, RR = 1.19, CI: 1.11–1.28) and in only one city (Rotterdam, the Netherlands) for women (RR = 1.09, CI: 1.03–1.14).

#### 3.2.2. Within-Country Studies

There were four “within-country” studies that considered all injuries aggregated, of which two from Sweden [47,49], one from Scotland [48] and one from Spain [50]. All report mortality differences; that of Bagher et al. [47] hospitalization for major trauma and that of Corfield et al. [48] includes data on emergency department attendances. Four social measures were put into relation with fatal and major trauma in the Bagher et al. [47] study in Malmö, Sweden: social assistance within the household, level of education, income and capital income are examined. Matching cases and controls for age-, gender- and district, they found that low income and social assistance within the household were both associated with major trauma but not the educational level. For the whole of Scotland, using an area level deprivation index consisting of 10 deciles, Corfield et al. [48] observed that the incidence of emergency department attendances for injuries but not that of in-hospital case fatalities due to injuries increased significantly with deprivation (least deprived IRR = 0.43, CI: 0.32–0.58 compared to most deprived IRR = 189.2, CI 180.6–197.8). In Sweden, the study on male adults 40–57 years conducted by Falkstedt et al. [49] studied whether psychosocial functioning and intelligence contribute to explain socioeconomic inequalities in a range of causes of premature death, including all injuries considering different measures of social position. They found that socioeconomic inequalities in injury mortality by education, occupational class and income was attenuated with 46% (CI: 28–65%), 52% (CI: 35–68%) and 38% (CI: 24–50%) respectively after adjusting for both individual psychosocial functioning and intelligence [49]. Finally, in Madrid, Spain Zoni et al. [50] used an ecological design found a statistically positive association between area deprivation and injury mortality in all age groups (<15, 15–44, 45–74, >75 years) with the largest differences for women in the age group 15–44 years (IRR = 1.52, CI: 1.49–1.55) and for men among those less than 15 years (IRR = 1.49, CI: 1.45–1.52).

From among the four studies dealing with road traffic injuries within countries, two are from England [51,54], one from Belgium [52], and one emphasized the social distribution of non-use of car seats as a risk factor across Slovenia [53]. All but the latter were ecological studies. Looking at the income based (two categories) distribution of road traffic injury mortality among men and women from Flanders, Pirdavani et al. [52] found a negative association between income level and casualties, with significance varying based on road user type and sex. In England, at country level and for the period 2010–2011, Hughes et al. [51] found almost threefold odds of emergency department attendances for (any) road traffic injury in children aged 0-14 years from most deprived areas compared to those from least deprived ones (adjusted OR 2.77; *p* < 0.001). In London more specifically, for the period 2000–2009, Steinbach et al. [54] investigated whether the social distribution of pedestrian injuries from casualties and collisions differed when paying attention to three aspects: (1) comparing all hours of the day vs. weekday morning commuting hours; (2) adjusting for the road environment (e.g., road density, traffic flow, speed); and (3) considering the child ethnic group ‘Black’, ‘White’ or ‘Asian’. For “all hours of the day” they observed that injury rate ratios increased with increasing area deprivation for the ‘White’ and ‘Asian’ children, but no such association for ‘Black’ children [54]. During the morning commute, injury rates were lower for all three ethnicity groups compared to all hours of the day, but still highest for ‘Black’ children whose rates tended to decrease with increasing levels of deprivation. After adjustment for the road environment, rate ratios in less deprived areas for all hours were substantially reduced for ‘White’ and ‘Asian’ children but remained the same in ‘Black’ children. During the time of the morning commute, the rates were similar across the deprivation quintiles for ‘White’ and ‘Asian’ children and substantially reduced for ‘Black’ children in the least deprived areas but increased for those in the most deprived area [54]. Finally, considering several measures of parental socioeconomic status (parental education by sex, family material welfare, and SES of area of residence) in relation to the use/non-use of car seats for their children among male and female parents, Rok-Simon and colleagues [53] found that two factors were associated with higher odds of non-use of child car seat: parental education, vocational or less compared to university (with a higher OR among mothers than fathers; OR 4.8 compared to OR 2.60 respectively) and SES of area of residence, poor (OR 1.37) or medium area (OR 2.37) compared to good.

The one study on (absolute and relative) social inequalities in falls described in the last part of Table 3 [55] is from Skåne, Sweden and concerns sex-specific injury mortality among older adults ages 50–75 years and education level (3 categories). Significant education-based associations are found both in absolute (SII) and relative (RII) terms in favour of well-educated men (SII 15.5; RII 2.19), but no similar associations are found in women. In total, 34% of men’s falls deaths were attributable to lower education [55].

Finally, two of the three studies on burns and poisonings are contributions from England [51,56], both are ecological, and the third one is from Spain [50] at the individual level but having social position expressed in terms of area deprivation. Considering five age groups of children and young adults and adjusting for age and sex, Baker et al. [56] found a two-fold higher rate of poisonings and 50% higher rate of burns leading to hospitalization or death when comparing 0–24-years old in the most deprived versus least deprived quintile; the steepest socioeconomic gradient was found for poisonings among 20–24-year olds (adj IRR = 2.63; CIs = 2.24–3.09). Focusing on paediatric emergency attendances, Hughes et al. [51] found elevated odds for children from the most deprived areas compared to least deprived for both poisonings (adj OR = 2.84; *p* < 0.001) and burns (adj OR = 2.14; *p* < 0.001). Similar associations were found in the case of Madrid residents (Spain) considering male and female separately and four age categories (<15, 15–44, 45–74, 75+) [50]; for burns, the incidence of injury rate ratios (IRRs) were highest in those from the most deprived living area compared to those from the least deprived areas and the largest differences between the lowest and highest quintiles were found in girls less than 15 years for burns (IRR = 1.89, CI: 1.65–2.18) and for poisonings (IRR = 2.08, CI: 1.48–2.94). In males, the largest differences between quintiles for burns were in the age group 15–44 years (IRR = 1.73, CI: 1.56–1.92) and in poisonings among men ages 45–74 (IRR = 1.80, CI: 1.34–2.41) [50].

## 4. Discussion

### 4.1. Changes over Time in Inequalities in Injury

Studies reporting changes in social differentials over time are quite heterogeneous, from the question they address to the nature of the outcome (e.g., injury cause or severity) and the social measure considered. While the results from the cross-country studies are in line with those already reported in recent decades, no obvious pattern emerges from the within-country literature, which comes from very few countries of the Region. In fact, both cross-country studies [30,31], using very similar data on childhood injury mortality, show expected and substantial global mortality decreases but demonstrate that the gains came at the expense of widening social inequalities in childhood injuries between HICs and LMICs echoes. The latter echoes the findings of a child review from within-country studies dealing with less recent data [28]. Increasing differences between HICs and LMICs in the WHO European Region have been attributed to more rapid pastes of decrease in HICs, more strikingly so for road traffic injuries which echoes previous European country-specific findings [57].

Within-country studies on adults or on general populations are from Spain (regional comparisons) [33] and Norway [35] and Finland [32] (individual-based comparisons). In Spain, comparisons of all injury mortality between the poorest and the richest provinces did not show any difference over time. In Norway, relative differences in injury mortality among older adults narrowed between education groups whereas in Finland, relative occupation-based differences in long-term injury-related sickness absence narrowed but absolute ones did not quite change. For children, a child injury study from the United Kingdom did not find any income gradient at any point in time when considering children sustaining several injuries [34]. Yet, as this group of children is a very small and particular segment of the population, it can definitely not be inferred from the finding that social inequalities in childhood injuries do not exist. In fact, numerous previous studies from the UK and the one using recent data presented in this review provide evidence of persisting injury inequalities [58]. Even cause-specific studies from the UK covered in this review point in that direction, showing rates of specific injuries like burns and poisonings decreasing in small children but socioeconomic inequalities persisting and also rates of poisonings increasing and (strong) socioeconomic gradients remaining among adolescents.

Explanations to those mixed results are many and include among others the study context and the study group (i.e., depending on the geographic area and population group under study), but also the social measure considered and the “homogeneity” of the outcomes considered. Injury mortality studies based on “all injuries” incorporate wide-ranging events, intentional and unintentional, that may have a very different genesis; the same applies to studies based on all unintentional injuries that actually incorporate many different causes. When the outcome is very heterogeneous studies may not be sensitive enough to identify, beside overarching changes in risk levels, time variations in how social groups compare to one another. There may even be scenarios that go in different directions—e.g., increases among social groups for some causes and decreases for others. Furthermore, a recent review of disparities in non-fatal injury outcomes found limited evidence of socially patterned inequities; the authors point to selection or retention biases being key limiting factors, as well as the focus on variations in outcomes only by sex [13]. These issues may explain the scarcity of studies on injury interventions identified in this review.

Road traffic injuries, for instance, relate to a far more specific environmental context but the categories still incorporate various categories of road users—pedestrians, bicyclists, motorcyclists, etc. The country-specific studies from Israel [36] and the UK [37,38,39] both report downward trends in levels of injuries and persisting inequalities. In Israel, in injury mortality and morbidity between Arabs and Jews (all ages) and in London, in child pedestrian injuries among “Black”, “White” and “Asian” and the magnitude of the reduction was similar across groups.

### 4.2. Studies in Inequality in Injuries Not Investigating Changes over Time

The cross-country studies making use of relatively more recent data are heterogeneous in the population group considered and the approach chosen. They indicate that nationwide country-level characteristics like income inequality or economic level are associated with injury mortality in children (all injury and traffic) to the detriment of those from countries with higher income disparities and lower income level [45,46]. This is much in line with previous evidence concerning children specifically [2] or other age groups from the Region [59] and globally [60]. When focus is placed on intra-regional deprivation levels across countries, all injury-mortality differentials (all ages) are common but yet not consistent, and they are more frequent in men than in women [43]. When men and women injury mortality is compared on the basis of education level, some differences are found in adult men but not adult women. These results are robust after controlling for many risk factors [42]. For road traffic injury and for fall mortality, the patterning is somewhat changed. Associations with education are found for road traffic injuries among women (to the detriment of the low-educated) [44] and with area deprivation among men [43]. In the case of falls, associations were uncommon but when found, they were more often with education [44] than area-based deprivation [43], and similar for both sexes [44]. This association has been observed in previous empirical studies, some which are highlighted in the previous review of European studies on inequality in injuries [44]. Using education as a measure of socioeconomic measure in older adults has the value of being stable as it is normally completed early in adulthood [44]. As an exposure, it captures a range of aspects pertaining to level of “understanding” or “knowledge” as well as links to physical “fitness”. Older people who are less educated have, in general, poorer health and are frailer than their more educated peers [61], and are therefore more at risk of both falling and being injured as a result of a fall.

Looking at the within-country studies which are also from a limited number of countries, the three all-injury studies (Sweden, Spain, Scotland) yield mixed results. In comparison, cause-specific studies (road-traffic injuries, burns and poisoning) tend to be more consistent. The road-traffic injury studies from Belgium and England point to inequalities to the detriment of those from less well-off living areas: in Flanders, for injury mortality among men and women from different categories of road users [52]; in England with an increase in injuries requiring emergency department attendances from areas of high deprivation [51], and in London variations in pedestrian injuries among “White”, “Asian” and “Black” children [54]. For the very young, a parental behavioural study revealed higher odds of non-use of child car seat among parents less educated or coming from less deprived areas [53]. Similarly, for burns and poisonings studies from England (area-based) [51,56] and Spain (individual living area) [50] report large differences for different injury severity levels and for several age groups, from childhood and adolescence (England and Spain) up to adulthood (Spain).

### 4.3. Physical Environment in Inequalities in Injuries

Environmental interventions for injury control and prevention have for long been promoted for their ability to reduce not only injury risks [62] but also social inequalities in injuries [63,64]. This is true for enhanced safety in the road traffic environment and elsewhere [65,66]. In this review, very few studies actually considered environmental aspects except for two concerned with road traffic injuries. Two modifications of the physical environment were evaluated for their potential to affect the social distribution of injuries: the extension of a highway in Glasgow, Scotland [37] and the imposition of 20 mph traffic speed zones on some London roads [37]. The highway extension had no effect on the social distribution of road traffic injuries as crashes tended to be located in the city centre, on non-highway roads in the city and in deprived areas [37]. By contrast, 20 mph zones decreased the number of road casualties similarly across quintiles of deprivation in London, but because of the higher number of road traffic casualties in deprived areas, social inequalities widened over time [38].

As for the home environment, the fact that adverse housing conditions appeared to be a likely pathway in the country-level association between income inequality, economic level and child injury mortality suggests that material deprivation, a fundamental cause of health inequalities [10], contributes to inequalities in child injuries in the Region [45,46]. Also, the studies support the notion that improvements in living standards may help not only in reducing injury risks for all children, but also in the reduction of disparities among them [65,66,67]. This is in line with a review of reviews showing that housing improvements can positively impact health inequalities for all ages [68]. Interventions to reduce injury risks associated with environmental hazards require population-based policies and regulations focusing on intersectoral approaches [69].

### 4.4. Strengths and Limitations

This is the first study that systematically reviews the state of knowledge on recent changes over time in injury in the WHO European Region, complemented by more recent studies dating from 2010 to 2018. The review had a broad scope in that it covered all injuries aggregated, in line with the studies showing reductions in global injury rates over time, as well as a variety of injury causes where rate reductions over time could be anticipated but social inequalities were not well documented. It also encompassed cross-country and within-country studies as they bring different perspectives to our understanding of social inequalities in injury—and health. For within-country studies, we used the established PROGRESS-Plus criteria to select relevant studies at the individual level, allowing for a range of measures to be taken into consideration. Also, the PRISMA guidelines were rigorously followed for selection of the articles (from three major databases) and extraction of the data.

A limitation of this review is that the studies reviewed were not individually assessed for their quality. Although we retained only studies that assessed the significance of the differences reported, we did not go further in our assessment of their methodological rigor and potential bias. This shortcoming is due in part to the heterogeneity of the study designs which require that different assessment tools be used. To overcome this, we attempted to use the Mixed Methods Appraisal Tool (MMAT) [70] in which all studies rated positively on the seven criteria, indicating an acceptable quality level. Yet the tool was not able to differentiate in quality between the studies as they all rated approximately the same. An additional potential shortcoming is the language restriction to English, implying that peer-reviewed empirical studies published in other languages were less likely to be captured. We believe that this shortcoming may be more towards a loss of data from the grey literature, which is not the focus of this systematic review. Finally, due to the wide heterogeneity of the studies, summarizing the findings in the form of meta-analysis was not feasible.

## 5. Conclusions

Studies on changes over time, both cross-country and within-country, reveal quite substantial downward trends in injuries in recent decades. This is valid for all-cause injuries as well as cause-specific injuries. However, social inequalities in injuries persist between countries. This is also found in most within-country studies although those countries contributing data are a handful and mainly from the North of Europe. The remaining studies investigate diverse questions in quite different population groups. Depending on the social measure and injury outcome considered, many report inequalities in injuries albeit to a varying degree. Despite the downward trends in risk levels, relative social inequalities in injuries remain a persisting public health issue in the Region.

## Figures and Tables

**Figure 1 ijerph-16-00653-f001:**
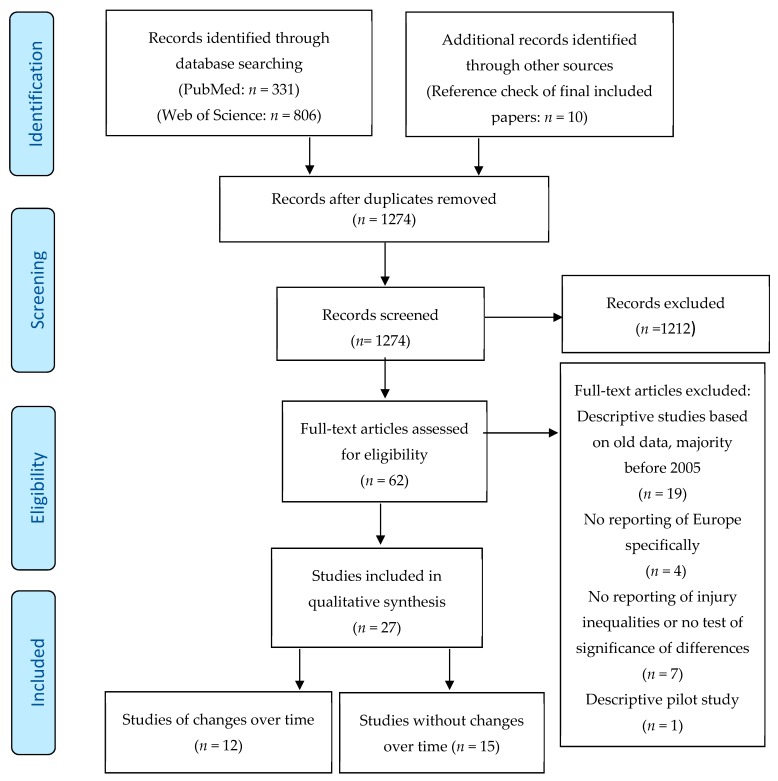
Preferred Reporting Items for Systematic Reviews and Meta-Analysis (PRISMA) flow diagram of studies on injury-related inequalities.

**Table 1 ijerph-16-00653-t001:** Search terms used for the search strategy.

(“sociological factors”[MeSH Terms] OR disadvantaged[All Fields] OR disadvantage[All Fields] OR deprived[All Fields] OR social[All Fields] OR socio*[All Fields] OR sociological[All Fields] OR “vulnerable populations”[MeSH Terms] OR vulnerable[All Fields] OR vulnerability[ALL Fields] OR “psychosocial deprivation”[MeSH Terms] OR psychosocial[All Fields] OR psycho-social[All Fields] OR “socioeconomic factors”[MeSH Terms] OR socioeconomic[ALL Fields] OR socioeconomic[ALL Fields] OR deprivation[All Fields] OR sociodemographic[All Fields] OR socio-demographic[All Fields])
AND
(“injuries”[MeSH Terms] OR “injury”[MeSH Terms] “accidents”[MeSH Terms] OR “injuries”[Title/Abstract] OR “injury”[Title/Abstract] OR “accidents”[Title/Abstract] OR “falls”[Title/Abstract] OR “poisoning”[Title/Abstract]) OR “drowning”[Title/Abstract]
AND
(inequality[Title/Abstract] OR inequity[Title/Abstract] OR inequities[Title/Abstract] OR inequalities[Title/Abstract] OR unequal[Title/Abstract] OR “environmental justice”[Title/Abstract] OR “environmental injustice”[Title/Abstract])
AND
(“2010/01/01”[Date-Publication]: “2018/11/31”[Date-Publication])

**Table 2 ijerph-16-00653-t002:** Descriptive characteristics of studies on injury inequalities investigating changes over time, cross-country and within-country.

Author/Year	Study Design/Reference Period	Setting/Population	Age Group, Sex	Social Measure(s)	Measure of Association	Injury Outcome	Results
**Cross-country**							
Göpfert et al., 2015 [31]	Ecological, 2000 vs. 2011	WHO European Region, 53 countries	Children 1–14 years	Gross national income	Mortality rate ratios between HIC and LMIC	All injuries split by cause (6 categories of unintentional injuries, 3 categories of intentional injuries)	All injuries: Relative inequalities widened between LMIC and HIC from 2000 to 2011 and mortality rate ratio between the two income groups increased by 31% from 4.31 in 2000 to 5.64 in 2011 (*p* < 0.001); variation by injury cause. Unintentional injuries: Inequalities widened between two income groups; mortality rate ratio increased from 4.36 to 6.04 (*p* < 0.001) despite decreases in number of deaths and mortality rates; rate ratio increased 94% for road traffic injuries (1.96 to 3.8, *p* < 0.001)
Sethi et al., 2017 [8]	Ecological, 2000 vs. 2015	WHO European Region, 53 countries	Children 1–14 years	Gross national income	Mortality rate ratios between HIC and LMIC	All injuries split by cause (6 categories of unintentional injuries, 3 categories of intentional injuries)	All injuries: Mortality rates persistently higher in LMIC compared to HIC between 2000 and 2015 and mortality declines greater in HIC than in LMIC; increased rate ratio between LMIC and HIC from 4.75 (CI 4.62–4.89) in year 2000 to 6.21 (CI 5.95–6.49) in year 2015 (31% increase, *p* < 0.001, ratio 1.31) Unintentional injuries: Rate ratio increased by 41% (*p* < 0.001) due to faster decline in HIC mortality from road traffic injuries (130% higher, *p* < 0.001)
**Within countries**							
**All injuries:**							
Pekkala et al., 2017 [32]	Cohort, 2005–2014	Finland	Adults 25–64 years, stratified by sex	Occupation, 3 classes	Slope Index of Inequality (SII) and Relative Index of Inequality (RII)	Long-term sickness absence due to all injuries	No significant change in sickness absence due to injuries for absolute differences but relative differences narrowed over time in men (*p* < 0.0001) due to a strong decline in prevalence among manual workers
Regidor et al., 2014 [33]	Ecological, 1970–2010	Spain, province level	All ages	Area of residence, provincial income	Absolute and relative differences (ratios)	All injury mortality	No association between absolute and relative quintiles of provincial income and premature injury mortality until 2010 (ratio of poorest province versus richest province 1.24, *p* < 0.05)
Shackleton et al., 2016 [34]	Cohort, 1980 vs. 2012	United Kingdom	Children 9–13 years	Household income, 3 levels	Odds ratio per year; Interaction income group and cohort	2 or more injuries requiring medical attention	No significant change 1980 (OR 1.09, CI: 0.95–1.25) vs. 2012 (OR 1.23, CI: 1.02–1.47) in odds of being injured when comparing lowest and highest income group; change over time (income × cohort), *p* = 0.18
Strand et al., 2014 [35]	Cohort, 1960–2010	Norway	Adults 45–74 years, stratified by sex	Education, 3 levels	Slope Index of Inequality (SII) and Relative Index of Inequality (RII)	All injury mortality	Injury mortality rates consistently higher in both male and female adults with basic versus higher education, persisting over time (SII 61 in men, 20 in women in 2010)
**Road traffic injuries:**							
Magid et al., 2015 [36]	Time series, 2003–2011	Israel *	All ages	Ethnicity: Arab and Jew	Injury and mortality rate ratios (RR)	All road traffic injuries	Mortality—reduction over time in both groups but at greater pace among Jews; Arabs consistently and increasingly at higher risk; RR 8.81 for 0–4 years, RR 3.08, 60–64 years Morbidity—reduction over time in both groups at a similar pace; Arabs consistently at higher risk
Olsen et al., 2017 [37]	Observational, 2008–2014	Glasgow, Scotland	All ages	Area level (Scottish Index of Multiple Deprivation), 5 quintiles	Annual count of clustered injuries (fatal, serious, slight)	All road traffic injuries	Majority of clustered injuries occurred in the three most deprived areas (annual proportion 30.6 in most deprived versus 1.0 in the least deprived for most recent year 2014); new motorway had no impact on the socioeconomic patterning
Steinbach et al., 2011 [38]	Observational, 1987–2006	London, England, 33 boroughs	All ages	Area level (Index of Multiple Deprivation), 5 quintiles	Trend across quintiles in (1) average annual percentage decline and (2) percentage reduction after 20 mph zones	All road traffic injuries and casualties; Killed and severely injured (KSI); Pedestrian casualties	All casualties: Graded decline over time from 1.2 in most deprived to 2.5 in least (*p* < 0.0001); no significant difference in % reduction after 20 mph zones (*p* = 0.62) Pedestrian/all casualties: Graded decline over time from 3.2 in most deprived to 4.0 in least (*p* < 0.015); no significant difference in % reduction after 20 mph zones (*p* = 0.60) KSI: No sig difference across quintiles either in decline over time or in % reduction after 20 mph zones
Steinbach et al., 2014 [39]	Ecological, 2001 versus 2011	London, England, 33 boroughs	Children 5–9 and 10–14 years	Area level deprivation (Index of Multiple Deprivation), 10 deciles; area ethnic population density (3 categories); child ethnicity, 3 categories	Incidence of injury rate ratios (IRR) accounting for characteristics of the road environment (density of roads and junctions, speed, traffic volume)	Pedestrian injuries	Significantly reduced (*p* < 0.001) risk of child pedestrian injuries by more than half in all three ethnic groups over time: ‘White’ children IRR 0.488; CI: 0.45–0.53 ‘Asian’ children IRR 0.420; CI: 0.37–0.48 ‘Black’ children IRR 0.489; CI: 0.45–0.53
**Other injuries:**							
Orton et al., 2014 [40]	Cohort, Four 5-year periods from 1990–2009	United Kingdom	Children 0–4 years	Area level deprivation (Townsend Index), 5 quintiles	Injury incidence ratios (IRRs) and attributable risk fraction	Injury treated by a general practitioner for burns and poisonings	Poisonings and burns significantly decreased over time (IRR test for trend *p* < 0.001) for all quintiles but consistently higher incidence with increasing deprivation, even when adjusting for sex, age and area deprivation; 28% of poisonings and 30% of burns attributable to deprivation
Tyrrell et al., 2016 [41]	Cohort, 1992–2012	United Kingdom	Children 10–17 years	Area level deprivation (Townsend Index), 5 quintiles	Adjusted incidence of injury rate ratios (aIRR)	General practice visits for poisonings	Significant positive association between poisoning incidence and deprivation which remained consistent over time; (aIRR 2.83, CI: 2.34–3.40 in 1992–1996 and 2.63, CI: 2.41–2.88 in 2007–2012)

* Israel is a member state of the WHO Regional Office for Europe.

**Table 3 ijerph-16-00653-t003:** Descriptive characteristics of studies on injury inequalities not investigating changes over time, cross-country, and within-country.

Author/Year/Reference	Study Design/Reference Year for Injury Outcome Data	Setting/Population	Age Group/Sex	Social Measure(s)	Measure of Association	Injury Outcome	Results
**Cross-country**							
Gallo et al., 2012 [42]	Cohort, 2002–2006	9 European countries, cities/national/regional	Adults 40–65 years	Education, 4 levels	Relative Index of Inequality (RII)	Mortality—All injuries	Men: Injury mortality significantly lower for those with highest education than lowest education (IIR = 0.56, CI: 0.35–0.90 crude); not significantly affected by adjustment for one or multiple risk factors (smoking status, alcohol consumption, leisure physical activity, fruit and vegetable consumption, body mass index (IRR = 0.61, CI: 0.38–0.98 adjusted) Women: No association between education level and injury mortality (IRR = 1.28, CI: 0.67–2.45) and unchanged when adjusting for one or multiple risk factors (IRR = 1.19, CI: 0.61–2.30)
Gotsens et al., 2013 [43]	Ecological, 2000–2009	15 European countries, cities and regions	All ages in five-year groups, stratified by sex	Index of socioeconomic deprivation based on 5 measures, 3 quintiles	Relative risk	Mortality—All injuries	Men: Positive association in majority of northern and western cities; no association in majority of eastern European cities; highest differences in Stockholm (RR = 1.27, CI: 1.22–1.31) Women: Positive association in 7 cities; no association in southern cities and no majority association in central eastern cities; highest differences in Stockholm (RR = 1.17, CI: 1.12–1.23)
						Mortality—Road traffic injuries	Men: Significant positive association in six cities; highest differences in Stockholm (RR = 1.18, CI: 1.07–1.29) Women: No association in cities studied
						Mortality—Falls	Men: Positive association in 4 cities; highest differences in Lisbon, Southern Europe (RR = 1.19, CI: 1.11–1.28) Women: Positive association only in Rotterdam, Western Europe (RR = 1.09, CI: 1.03–1.14)
Mackenbach et al., 2014 [44]	Mixed cross-sectional and longitudinal, varying from 1998–2007, country-specific	16 European countries, national/regional/city	Adults 30–79 years, stratified by sex	Education, 3 levels and results ‘low’ vs. ‘high’	Relative risk (RR) at 95%, 99% and median confidence intervals	Mortality—Road traffic, all users Mortality—Falls	Higher mortality risk among the ‘low’ vs. ‘high’ educated, more so for men (median RR 2.06) than for women (median RR 1.26) Higher mortality risk among the ‘low’ vs. ‘high’ educated, similar for both sexes (median RR 1.84 men, median RR 1.55 women)
Sengoelge et al., 2013 [45]	Ecological, 2006	26 European countries	Children 1–14 years	-Economic level, 2 measures -Living conditions, 2 indexes (housing strain, neighborhood strain)	Standard mortality ratios and correlations for association measures	Mortality—All injuries	Economic level: Significant correlation between income inequality and mortality (r = 2.05, CI: 1.07–3.03) and between GDP and mortality (r = −6.55, CI: −9.31–−3.80) Living conditions: Significant correlation between housing strain and mortality (r = 5.94, CI: 1.58–10.30); no correlation between neighborhood strain and mortality (r = 5.67, CI: −2.32–13.67) Economic level + living conditions: Income inequality and housing strain (r = 1.77, CI: 0.51–3.04) and GDP and housing strain (r = −6.02, CI: −9.61–−2.42) partially modified the correlation with mortality
						Mortality—Road traffic, all users	Economic level: Significant correlation between income inequality and mortality (r = 0.61, CI: 0.27–0.95) and between GDP and mortality (r = −2.50; CI: −3.23–−1.77) Living conditions: Significant correlation between housing strain and mortality (r = 2.31; CI: 0.98–3.64) Economic level + living conditions: No significant modification in correlation between income inequality + housing strain and mortality (r = 0.39; CI: −0.03–0.81); sign. partial modification GDP+ housing strain and mortality (r = −2.25; CI: −3.20–−1.31)
Sengoelge et al., 2014 [46]	Ecological, 2006	26 European countries	Children 1–14 years in 3 age categories (1–4, 5–9, 10–14)	-Income inequality (80:20 ratio) -2 indexes of living conditions: housing and neighbourhood strain	Standard mortality ratios and correlations for association measures	Mortality—All injuries	Significant correlation between income inequality and mortality (r = 0.70, *p* ≤ 0.001) and housing strain and mortality (r = 0.46, *p* = 0.017) but not significantly between neighbourhood strain and mortality (r = 0.24, *p* = 0.239); very small age-specific differences
**Within countries**							
**All injuries:**							
Bagher et al., 2016 [47]	Case-control, 2011–2013	Malmö, Sweden, 10 districts	3 age groups (all ages, 25+ years, 18+ years)	Five measures: -education, 3 levels -income, 3 levels and capital income -recipient of household social assistance	Odds ratios (OR)	Hospitalisation for major trauma (New Injury Severity Score > 15 or death at trauma scene sent for autopsy)	Increased odds of major trauma in relation to no income in 18+ years (OR = 1.6, CI: 1.0–2.4) and social assistance in all ages (OR = 2.3, CI: 1.3–4.1) No increased odds of major trauma in relation to low vs. medium/high education (OR = 1.3, CI: 0.8–2.2)
Corfield et al., 2016 [48]	Cohort, 2011–2012	Scotland	Adults 16+ years in 8 age categories, stratified by sex	Area level deprivation (Scottish Index of Multiple Deprivation), 10 deciles	Incidence of injury rate ratios (IRRs) and odds ratios (OR)	In-hospital case fatality and emergency department attendances for trauma	In-hospital injury case fatality: odds not associated with socioeconomic deprivation when adjusted for age and sex Emergency department attendances for injuries: increased with increasing deprivation with patients in most deprived decile having a 2.5-fold higher incidence of trauma vs. patients in least deprived decile (IRR = 0.43, CI: 0.32–0.58 least deprived vs. IRR = 189.2, CI: 180.6–197.8 most deprived)
Falkstedt et al., 2013 [49]	Cohort, 1991–2008	Sweden	Adults 40–57 years, males only	Three measures: -Education, 5 levels -Occupation, 5 classes -Income, 5 quintiles	Relative index of inequality (RII)	Mortality	Lower scores of intelligence and lower psychological functioning associated with higher injury mortality Socioeconomic inequalities in injury mortality by education, occupational class and income attenuated with 46% (CI: 28–65%), 52% (CI: 35–68%) and 38% (CI: 24–50%) respectively after adjusting for both individual psychosocial functioning and intelligence
Zoni et al., 2017 [50]	Cross-sectional, 2012	Madrid, Spain	All ages in 4 age groups (<15, 15–44, 45–74, >75), stratified by sex	Area level deprivation, 5 quintiles	Incidence of injury rate ratios (IRR)	Mortality	Statistically significant higher injury mortality incidence with increasing deprivation in all age groups; largest differences for women 15–44 (IRR = 1.52, CI: 1.49–1.55) and men aged <15 (IRR = 1.49, CI: 1.45–1.52)
**Road traffic injuries:**							
Hughes et al., 2014 [51]	Ecological, 2010–2011	England	Children 0–14 years	Area level deprivation (Index of Multiple Deprivation), 5 quintiles	Adjusted odds ratios (AOR)	Emergency department attendances for road traffic injuries	Approximately 3-fold higher odds of emergency department attendances in those from most deprived vs. least deprived (AOR = 2.77, *p* < 0.001) and relationship similar after adjusting for age and sex
Pirdavani et al., 2017 [52]	Ecological, 2010–2012	Flanders, Belgium	Adults, stratified by sex	Income (average household aggregated to area), 2 levels	Parameter estimates with standard error, z-value	Mortality and injury casualties, 3 types	Negative association between income level and casualties with significance varying based on road user type and sex: significant predictor of male car driver injury crashes but not for female car drivers; significant predictor of both male and female car passenger injury crashes; not significant predictor of male or female pedestrian and cyclist injury crashes
Rok-Simon et al., 2017 [53]	Cross-sectional survey, 2015	Slovenia	Adults, stratified by sex	3 measures: - Parental education, 4 levels - Family material welfare, 3 levels - SES of area of residence (income tax base per capita at municipality), 3 levels	Odd ratios (OR)	Non-use of child car seat	Measures associated with higher odds of non-use of child car seat: -parental education for both men and women when comparing vocational/less to university (OR = 4.08, CI: 2.18–7.62 mother and OR = 2.60, CI: 1.40–4.83 father) -SES of area of residence area, poor (OR = 1.72, CI: 1.10–2.71) or medium (OR = 2.37, CI: 1.47–3.82) vs. good Measures not associated: family material welfare
Steinbach et al., 2014 [54]	Ecological, 2000–2009	London, England, 33 boroughs	Children 5–9 and 10–14 years	2 measures: -Area level deprivation (Index of Multiple Deprivation), 10 quintiles -Ethnicity, 3 categories	Incidence of injury rate ratios (IRR) accounting for exposure	Pedestrian injuries (casualties and collisions)	Increase in injuries with increasing deciles of deprivation for ‘Whites’ and ‘Asians’ after adjusting for the road environment, but no difference in rates for ‘Black’ children at all hours Morning commute: controlling for the road environment substantially reduced rate ratios for ‘White’, ‘Asian’ children and tendency for decreased rates with decreasing levels of deprivation for ‘Black’ children
**Falls:**							
Ahmad Kiadaliri et al., 2018 [55]	Cohort, 1998–2014	Skåne region, Sweden	Adults 50–75 years, stratified by sex	Education, 3 levels	Slope and relative indices of inequality (SII/RII),	Mortality from falls	Both SII and RII reveal statistically significant educational inequalities in falls mortality in men in favour of well educated (SII 15.5, RII 2.19); 34% falls deaths in men attributable to lower education, but not in women
**Burns and poisonings:**							
Baker et al., 2016 [56]	Cohort, 1998–2011	England	Children 0–24 years in 5 age categories	Area level deprivation (Index of Multiple Deprivation), 5 quintiles	Adjusted incidence rate ratios (aIRR)	Mortality and hospitalisations from poisonings and burns	Positive association between deprivation and poisoning (aIRR = 2.12, CI: 1.68–2.69) and deprivation and burns (aIRR = 1.53, CI: 1.40–1.68) incidence after adjusting for age and sex; steepest socioeconomic gradient found in poisonings among ages 20–24 (aIRR = 2.63, CI: 2.24–3.09)
Hughes et al., 2014 [51]	Ecological, 2010–2011	England	Children 0–14 years	Area level deprivation, Index of Multiple Deprivation, 5 quintiles	Adjusted odds ratios (AOR)	Emergency department attendances for poisonings and burns	Higher odds of emergency department attendances for children from most deprived areas compared to least deprived areas for poisonings (AOR 2.84) and burns (AOR 2.14) *p* < 0.001 after adjusting for age and sex
Zoni et al., 2017 [50]	Cross-sectional, 2012	Madrid, Spain	All ages in four age groups (<15, 15–44, 45–74, >75), stratified by sex	Area level deprivation, 5 quintiles	Incidence of injury rate ratios (IRR)	Primary care consultations for poisonings and burns	Poisonings: IRR higher for all ages in the lowest vs. highest SES with largest differences in girls less than 15 years (IRR = 2.08, CI: 1.48–2.94) and in men 45–74 (IRR = 1.80, CI: 1.34–2.41) Burns: IRR higher for all ages in the lowest vs. highest SES with largest differences for girls less than 15 years (IRR = 1.89, CI: 1.65–2.18) and men 15–44 (IRR = 1.73, CI: 1.56–1.92)

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
