# Peer review of "Are There Changes in Inequalities in Injuries? A Review of Evidence in the WHO European Region"

_ijerph, 2019, doi:10.3390/ijerph16040653_

Round 1

Reviewer 1 Report

This paper makes a valuable contribution to the field, and appears to be quite a comprehensive study. You have pointed out most of the paper's limitations yourselves, the most obvious being that there are so many injuries, jurisdictions, study sizes, causes, contributors, and other variables involved in the studies you look at in this meta-analysis, that it is somewhat hard to know what to draw from it. That said, that all of this data has been brought together in one document is useful, and it does allow for some interesting comparisons. It also points to the many places where more targeted studies are needed.

There are a few typos in the text but the language is excellent (note: correct date range in line 146). 

I have a specific question at lines 273-275: can you comment somewhere in the text about how men's falls were attributable to lower education? This does not seem to be an intuitive connection at all, and I wonder about the use of 'attributable' here - is this a correlation, or is the claim that there is actually a causal link?

Otherwise, I think you have sufficiently contextualised the studies and your findings. 

Author Response

Reviewer 1:

There are a few typos in the text but the language is excellent (note: correct date range in line 146). 

Author response: The reviewer expresses a concern about a date range that we are not able to identify in our version of line 146. We have reviewed date ranges throughout the article and did not identify an error.

I have a specific question at lines 273-275: can you comment somewhere in the text about how men's falls were attributable to lower education? This does not seem to be an intuitive connection at all, and I wonder about the use of 'attributable' here - is this a correlation, or is the claim that there is actually a causal link?

Author response: The association between fall injury and low education among adults and older people has been found is several previous empirical studies and it was also highlighted in the previous review of European studies on inequality in injuries (ref). Mackenbach and colleagues (2015) define the value of using education as a socioeconomic measure in older adults as it is “the most stable measure of socioeconomic position because it is normally completed early in adulthood, which also avoids reverse causation problems (i.e., health outcomes at older ages cannot change a person's level of education) (…).” 

In that population, it is not only an assessment of “knowledge” but also one of “frailty”: older less educated people have in general poorer health – and are frailer – that their well-educated peers, which exposed them to higher risks of both falling and being injured as a result of a fall. We have included this information in the revised version of the manuscript – see lines 383-389. Furthermore, the term “attributable” was the one used by the authors cited and it echoed well the measure of association that they used which is why we did not change it.

Reviewer 2 Report

This is a well written and organized article. However, some aspects need to be clarified, and minor changes are also required to correct stylistic and grammatical errors (e.g. too long sentences, missing words). -- Need to formulate a clear research question, or hypothesis, or aim (lines 45-47, 64-66). -- It would be useful to provide a definition of 'injury' and 'social inequality' in the Introduction or Methods section (2.2). Injuries are categorized as burns, falls, poisonings and road traffic accidents [line 78]. Why were these four kinds chosen? -- WHO 2009 review is mentioned briefly in text (lines 56, 75). If this is a key text, provide more detail on its contents and also list it in references? -- Studies are grouped into 'studies of changes over time' and 'other studies.' The latter is not a good category. Could 'other studies' be provided a better descriptive label? It is mentioned in a few places that these are more recent. This is not sufficient. Did 'other studies' cover social inequalities without considering changes over time? -- Why is a study of Israel utilized (Table 2)? -- In Fig. 1 (PRISMA flow diagram), there is a * against 'other studies.' Is this because 2 studies are mentioned twice? Would it be possible to place the Hughes et al. and Zoni et al. studies in one category, instead of repeating them (in Table 3)? -- Some sentences become too long and could be shortened to improve clarity (e.g lines 255-263, 351-355). -- Begin a new paragraph from line 399-400, so that limitations are discussed separately from strengths,

Author Response

Reviewer 2:

This is a well written and organized article. However, some aspects need to be clarified, and minor changes are also required to correct stylistic and grammatical errors (e.g. too long sentences, missing words).

Need to formulate a clear research question, or hypothesis, or aim (lines 45-47, 64-66).

Author response: The aim of the study is two-fold. The primary one is to shed light on the state of knowledge concerning changes over time in social inequality in injuries in the European Region. A secondary aim is to complement this description with an updated review of studies from the Region on social inequalities in injuries in injuries that are not concerned with changes over time but that present data more recent than those presented in the former review from Laflamme et al. (2009). The text has been edited on line 68-72 for improved clarity.

It would be useful to provide a definition of 'injury' and 'social inequality' in the Introduction or Methods section (2.2).

Author response: We have used a broad perspective on the issue of social inequalities and looked into differences at several possible levels of observation: individual, area and bigger geographic areas like regions and countries A definition of ‘injury’ has now been added in methods line 76-77 from Robertson LS. Injury epidemiology. 2nd Ed. New York: Oxford University Press, 1998:265.

Injuries are categorized as burns, falls, poisonings and road traffic accidents [line 78]. Why were these four kinds chosen?

Author response: Those four causes were chosen as they are major causes of injury mortality and morbidity and previous empirical studies as well as the previous WHO review on social inequalities in injury (Laflamme et al. 2009) have shown that they tend to be associated with inequalities. Text added on line 86-87 to explain this.

WHO 2009 review is mentioned briefly in text (lines 56, 75). If this is a key text, provide more detail on its contents and also list it in references?

Author response: More detail has been added in line 57-59 and the reference to this review was listed in the references as #12.

Studies are grouped into 'studies of changes over time' and 'other studies.' The latter is not a good category. Could 'other studies' be provided a better descriptive label? It is mentioned in a few places that these are more recent. This is not sufficient. Did 'other studies' cover social inequalities without considering changes over time?

Author response: We have debated this a few times among the authors and we found it was convenient to indicate in the methods section what was meant by that and then use the term “other studies” in the remaining of the text for brevity. As it seems preferable to be more explicit throughout the text based on this comment, we now use ‘studies not investigating changes over time’.

Why is a study of Israel utilized (Table 2)?

Author response: Israel is an associated state of the European Union and is therefore included in the WHO European Region countries. This explanation has now been added to Table 2 as a footnote and in the text on line 171-172.

In Fig. 1 (PRISMA flow diagram), there is a * against 'other studies.' Is this because 2 studies are mentioned twice?

Author response: The group ‘other studies’ has been renamed for clarity based on the comment above so the asterisk has been removed.

Would it be possible to place the Hughes et al. and Zoni et al. studies in one category, instead of repeating them (in Table 3)?

Author response: We considered this at an earlier stage but as the results are synthesized by cause, we found that the format we choose suited better the flow. When some studies are repeated (which is not frequent), we do indicate that in the text.  

Some sentences become too long and could be shortened to improve clarity (e.g lines 255-263, 351-355).

Author response: Changes made (lines 279-288, 378-379 under new formatting of this version to the revisions).

Begin a new paragraph from line 399-400, so that limitations are discussed separately from strengths.

Author response: Change made (line 434).